# Atypical Electrophysiological Indices of Eyes-Open and Eyes-Closed Resting-State in Children and Adolescents with ADHD and Autism

**DOI:** 10.3390/brainsci10050272

**Published:** 2020-05-01

**Authors:** Alessio Bellato, Iti Arora, Puja Kochhar, Chris Hollis, Madeleine J. Groom

**Affiliations:** 1Division of Psychiatry and Applied Psychology, School of Medicine, University of Nottingham, Institute of Mental Health, Innovation Park, Triumph Road, Nottingham NG7 2TU, UK; iti.arora@nottingham.ac.uk (I.A.); msapk6@exmail.nottingham.ac.uk (P.K.); chris.hollis@nottingham.ac.uk (C.H.); maddie.groom@nottingham.ac.uk (M.J.G.); 2NIHR MindTech Healthcare Technology Co-Operative, Institute of Mental Health, Innovation Park, Triumph Road, Nottingham NG7 2TU, UK

**Keywords:** ADHD, autism, resting-state, EEG, alpha, alpha reactivity

## Abstract

Investigating electrophysiological measures during resting-state might be useful to investigate brain functioning and responsivity in individuals under diagnostic assessment for attention deficit hyperactivity disorder (ADHD) and autism. EEG was recorded in 43 children with or without ADHD and autism, during a 4-min-long resting-state session which included an eyes-closed and an eyes-open condition. We calculated and analyzed occipital absolute and relative spectral power in the alpha frequency band (8–12 Hz), and alpha reactivity, conceptualized as the difference in alpha power between eyes-closed and eyes-open conditions. Alpha power was increased during eyes-closed compared to eyes-open resting-state. While absolute alpha power was reduced in children with autism, relative alpha power was reduced in children with ADHD, especially during the eyes-closed condition. Reduced relative alpha reactivity was mainly associated with lower IQ and not with ADHD or autism. Atypical brain functioning during resting-state seems differently associated with ADHD and autism, however further studies replicating these results are needed; we therefore suggest involving research groups worldwide by creating a shared and publicly available repository of resting-state EEG data collected in people with different psychological, psychiatric, or neurodevelopmental conditions, including ADHD and autism.

## 1. Introduction

Attention deficit hyperactivity disorder (ADHD) affects about 5% of children worldwide, and it is characterized by symptoms of inattention and hyperactivity/impulsivity, which are associated with difficulties in regulating behavior and reduced adaptive functioning [1]. The development of mature strategies of behavior and arousal regulation is supported by specialization of structural and functional connections between cortical, limbic, and brainstem systems [2] and it is affected by environmental factors [3]. Despite researchers’ efforts that have resulted in numerous published studies which highlighted atypical brain functioning in ADHD, the difficulty in translating important research findings into clinical practice, added to the widely recognized “replicability crisis” in psychiatry, psychology, and neuroscience [4], have indirectly highlighted the need to identify specific domains which are more likely to be associated with sets of clinical symptoms (both condition-specific and transdiagnostic). A domain which has been quite widely studied in ADHD is brain functioning at rest.

“Resting-state” is the term which refers to situations when a person is not involved in any mentally challenging or physical activity. During resting-state, attention is less focused on external sensory information [5] and more on internal information, memories and thoughts [6]. It has been widely demonstrated that resting-state EEG is predominantly characterized by oscillatory activity in the alpha frequency band (8–12 Hz), especially over occipital scalp regions and more markedly at rest with eyes closed [7]. Moreover, alpha oscillations have been proposed to reflect a general physiological property of neuronal cells and brain systems, and therefore to be involved in different cognitive functions [8] and to be associated with functioning of autonomic and vegetative arousal systems [9]. Associations between alpha oscillatory patterns and physiological arousal at rest, have in fact been demonstrated by studies showing that increased alpha power during eyes-closed resting-state is associated with reduced arousal. Increased alpha activity during eyes-closed resting-state seems in fact to reflect an “interoceptive” state, characterized by imagination and reduced activation in brain areas involved in processing of visual information [10,11]. Moreover, compared to resting-state at eyes open, electro-dermal activity is reduced, signaling reduced activation of the autonomic nervous system [12,13,14,15,16] and increased vigilance, facilitating processing of visual information during eyes-closed resting-state [17].

Different theoretical models and previous empirical studies have proposed that the autonomic nervous system might be under-active in people with ADHD and this might be associated with chronic states of hypo-arousal and reduced vigilance at rest (see [18] for a review), while hyperactivity and restlessness might be strategies adopted by individuals with ADHD to up-regulate arousal [19,20,21]. Autism, which is characterized by social communication and social interaction difficulties, restricted and repetitive behaviors, interests or activities [22]; has been frequently found to co-occur with ADHD [23]. Similarly to ADHD, autism is likely to be characterized by dysregulated arousal at rest, however more predominantly in the form of hyper-arousal (manuscript in preparation [24]). We therefore decided to specifically investigate alpha power and the difference in alpha power between eyes-open and eyes-closed resting-state (i.e., “alpha reactivity”) in a sample of children with clinical diagnoses of ADHD and autism, which are likely to affect these measures.

For example, Newson and Thiagarajan have carried out a review on resting-state EEG profiles across different psychiatric conditions, including ADHD and autism [25]; they found that relative alpha (i.e., alpha measured in relation with other frequency bands, including delta, theta, and beta) is likely to be atypically reduced in children with ADHD, especially during eyes-closed resting-state. Conversely, autism has been found associated with reduced absolute alpha oscillatory activity, more specifically during eyes-open resting-state [25]. Only one study [17] has measured the difference in alpha power between eyes-open and eyes-closed (i.e., alpha reactivity), and has found reduced alpha reactivity in children with ADHD. This finding might suggest that abnormalities in adapting to situations with different loads of sensory information are likely to characterize this condition; however, alpha reactivity was measured using absolute alpha power (and not relative) [17] and, to our knowledge, no other study has investigated relative alpha reactivity in ADHD before. Moreover, although the presence of co-occurring symptoms of autism in people with ADHD is likely to have an interactive effect, further impacting brain functioning and development [26], only few studies focused on investigating alpha oscillatory patterns in children with co-occurring ADHD + autism. Shephard et al. [27], for example, demonstrated that children presenting symptoms of autism (with or without ADHD) had reduced alpha power compared to those without autism (i.e., typically developing controls and children with ADHD-only). However, in that study [27], an eyes-closed resting-state condition (where atypical electrophysiological patterns have been specifically reported in ADHD) was not included, and no other study has ever investigated alpha reactivity in children with co-occurring ADHD + autism.

We carried out a secondary analysis on previously collected data [28], investigating electrophysiological patterns of alpha power and alpha reactivity in a mixed sample of children with ADHD and/or autism, and in different resting-state conditions (i.e., eyes-closed and eyes-open), which no previous study has done before. This work therefore originated from the fact that only few previous studies have focused on mixed samples of children with ADHD and autism and at the same time analyzed EEG alpha oscillatory patterns in different resting-state conditions (i.e., eyes-closed and eyes-open), including alpha reactivity as an outcome measure. The original study [28] reported results on neural mechanisms of gaze cues and face processing in children with ADHD and/or autism; in this work, we specifically focused on absolute and relative alpha power in the occipital scalp area. We predicted that there will be reduced relative alpha power in children with as compared to without ADHD, and we investigated the possibility that this group effect is greater during eyes-closed than eyes-open; conversely, we predicted that there will be reduced absolute alpha power in children with autism, and we also predicted to find reduced alpha reactivity in children with ADHD, indicating impaired ability to regulate brain states in response to a change in context. 

Overall, we partially confirmed these hypotheses, having found decreased absolute alpha power in children with autism and reduced relative alpha power in children with ADHD, especially during eyes-closed resting-state. However, we could not replicate the results by Fonseca and colleagues [17], since alpha reactivity was not found to be reduced in children with ADHD. As highlighted in the conclusive section of the present paper, we anticipate our willingness to create a shared and publicly available repository of resting-state EEG data, to promote collaborations between researchers and research teams worldwide, in order to replicate these findings and advance our knowledge about patterns of electrophysiological oscillatory activity during resting-state in ADHD.

## 2. Materials and Methods

### 2.1. Participants

This study is based on a secondary analysis of data previously collected by Groom and colleagues [28]. Children between 8 and 15 years were recruited for the original study, and they were diagnosed with or were undergoing assessment for ADHD or autism. Consensus diagnosis (based on DSM-IV criteria) was reached between two clinicians (child and adolescent psychiatrists) to confirm the presence or absence of clinical diagnoses of ADHD and autism. All participants with ADHD who were on stimulant medication were required to stop taking their medication 36 h prior to the EEG session. As reported in the original paper, “all procedures performed in studies involving human participants were in accordance with the ethical standards of the institutional and/or national research committee and with the 1964 Helsinki declaration and its later amendments or comparable ethical standards” ([28], p. 1507).

Data from 43 children, including 20 typically developing controls, 9 children with ADHD-only, 6 children with autism-only and 8 with comorbid ADHD + autism, were used for the present study and are publicly available at the following link: https://osf.io/azkhs/. The effects of ADHD- and autism-diagnoses on the main outcome measures were investigated by using binomial between-subjects factors that reflected the presence or absence of a clinical diagnosis of ADHD or autism in a child, namely an “ADHD-factor” (no-ADHD: 26 children; ADHD: 17 children) and an “autism-factor” (no-autism: 29 children; autism: 14 children). We therefore tested the presence of effects associated with a specific condition (ADHD or autism) and the comorbid presence of ADHD and autism, although the size of the comorbid group was small. Sample characteristics and significant between-groups differences on demographic and clinical variables, are presented in Table 1.

### 2.2. Procedures

Electroencephalography (EEG) was recorded during a 4-min resting-state session, after participants completed an attentional task (task-specific findings were reported in the original paper [28]). The resting-state session was comprised of two conditions, namely eyes-closed and eyes-open, lasting 2 min each. During the eyes-open condition, children were asked to stay still on a chair and look at a fixation cross on a computer monitor, positioned 60 cm from the child. Before the beginning of the eyes-closed condition, children were asked to close their eyes, stay still on the chair and relax.

EEG data were recorded using a Biosemi Active II system (Biosemi, Netherlands) with 128 silver/silver chloride (Ag/AgCl) electrodes positioned according to an extended 10–20 montage, using a 500-Hz sampling-rate and impedances under 10 kΩ. During recording, signals were referenced to a common mode sensor to the left of Cz. Additional electrodes were placed at the inner orbital ridge and the outer canthus of each eye to record eye movements and on the left and right mastoids to record other artefacts. Pre-processing of EEG data was performed with Brainstorm [29], which is documented and freely available for download online under the GNU general public license (http://neuroimage.usc.edu/brainstorm). Flat or noisy channels were rejected, before being re-referenced to the average of the remaining channels and being filtered with 0.5 Hz high-pass, 70 Hz low-pass and 50 Hz notch filters. Signal Space Projection (SSP; [30]) was used to remove ocular artefacts, while Independent Component Analysis (ICA) was used to identify and then remove further artefacts, before segmenting EEG data in 2-s long consecutive epochs. Each epoch was subjected to Fast Fourier Transformation (FFT) with 10% Hanning window, and then averaged across epochs. Spectral absolute power was computed for each electrode in the delta (0.5–3.5 Hz), theta (3.5–8 Hz), alpha (8–12 Hz), and beta (12.5–22 Hz) frequency bands. Absolute spectral alpha power was extracted for each condition (eyes-open and eyes-closed) by averaging alpha power at the Oz electrode and at two neighbor midline electrodes. Alpha power values were subjected to a log-transformation to normalize the distribution before carrying out further statistical analysis. After calculating absolute and relative alpha power (which is the spectral power in the alpha band as a proportion of overall power across the other frequencies), we calculated alpha reactivity by calculating the difference between eyes-open and eyes-closed resting-state in absolute and relative alpha power.

### 2.3. Analysis Plan

All the analyses were carried out in JASP [31], a software freely downloadable from https://jasp-stats.org/. Gender and IQ were included as covariates in the models. Bayesian statistical analyses have been used to test the main hypotheses of the present study. Bayesian statistic is a data-focused approach that investigates the probabilities of different hypotheses or models, investigating how much the data fit with each [32]. More specifically, the Bayes Factor (BF), which is derived from Bayesian analyses, represents how many times the observed data is likely to fit with an alternative hypothesis, compared to the null [33]. We used JASP [31] to calculate the Bayes Factor of different candidate models where different main factors and interactions between these factors have been considered. We therefore investigated, for each analysis reported in the “Results” section, which model was the ‘best’ in predicting the observed data, namely which one showed the highest Bayes Factor in comparison to the null model (i.e., the one where no specific factor was included). This has been done before investigating the specific effects of the factors included in the “best” predictive models. A Bayes Factor higher than 1 is usually interpreted as evidence in support of the alternative hypothesis (e.g., the presence of a predictive effect of a factor in the model), while a Bayes Factor lower than 1 indicates evidence in support of the null hypothesis (in this case, the absence of a predictive effect of a factor in the model). In addition, Bayes Factors between 0.33 and 3 are usually interpreted as inconclusive, so that evidence in support of the null or the alternative hypothesis is “anecdotal” [33]. Further information about Bayesian statistics and Bayesian ANOVA can be found at https://osf.io/f8krs/.

## 3. Results

We investigated electrophysiological oscillatory patterns of alpha activity and reactivity in children with ADHD or autism. Separate repeated measures Bayesian ANCOVAs were carried out on absolute and relative alpha spectral power measured at the central-occipital scalp region, which were compared across the two Conditions (within-subjects factor; 2-levels: eyes-closed and eyes-closed) and in relation to ADHD- and autism-diagnoses (between-groups factors; 2-levels: yes/no), controlling for gender and IQ. Further, two Bayesian ANCOVAs were carried out on alpha reactivity (based on absolute and relative power) with ADHD and autism as between-groups factors and controlling for the effects of gender and IQ.

### 3.1. Absolute Alpha Power

We found evidence that the model which included (Condition + autism + Gender) as factors, best predicted the data on absolute alpha power (BF_model_ = 6.415; error % = 3.635; R^2^ = 0.690). We found strong evidence for the presence of an effect of Condition on absolute alpha power (BF_inclusion_ = 1,916,000), indicating that absolute alpha power was increased during eyes-closed compared to eyes-open (BF_corrected_ = 2,093,000; Figure 1). Our results provided anecdotal evidence of the presence of a main effect of autism (BF_inclusion_ = 2.530), indicating that children with autism (autism-only and ADHD + autism) had reduced absolute alpha power, compared to children without autism (ADHD-only and typically developing controls) (BF_corrected_ = 3.896; Figure 2). We found evidence in support of the absence of differences on absolute alpha power in relation to ADHD (BF_inclusion_ = 0.653). Overall, these findings are in line with our primary hypotheses and suggest the presence of increased absolute alpha power during eyes-closed than eyes-open resting-state, and reduced absolute alpha power during resting-state in children with autism.

### 3.2. Relative Alpha Power

When investigating relative alpha power, the following model was the best to predict the data: (Condition + ADHD + Condition * ADHD; BF_model_ = 8.834; error % = 1.924; R^2^ = 0.637). We therefore further investigated each predictive factor or interaction, which was added to this model.

Similarly to absolute alpha power, we found strong evidence in support of the presence of an effect of Condition on relative alpha power (BF_inclusion_ = 25,979), so that relative alpha power was increased during eyes-closed compared to eyes-open resting-state (BF_corrected_ = 26,493; Figure 3). These results highlight how alpha power increased, in relation to power in other frequency bands, from eyes-open to eyes-closed resting-state.

We also found evidence in support of a predictive effect of ADHD (BF_inclusion_ = 6.951) and anecdotal evidence in support of the presence of an interaction ADHD * Condition (BF_inclusion_ = 1.792). More specifically, post-hoc tests on ADHD-factor revealed that relative alpha power was reduced in children with ADHD (ADHD-only and ADHD + autism) compared to children without ADHD (ASD-only and typically developing controls) (BF_corrected_ = 166.592; Figure 4). This is in line with our predictions, suggesting that atypically reduced relative alpha power might be specifically associated with ADHD. However, before fully interpreting these results, it would be important to follow up the ADHD * Condition interaction.

As suggested by van den Bergh and colleagues [34], we investigated the interaction Condition * ADHD by running two separate Bayesian independent-samples t-tests to compare relative alpha power during eyes-open and eyes-closed resting-state, between children with or without ADHD. Using the default Cauchy prior distribution (scale parameter = 0.707), we found evidence in support of the presence of a between-groups difference on relative alpha power during eyes-closed resting-state (BF_10_ = 19.002; error % < 0.001), and anecdotal evidence in support of the presence of a between-groups difference during eyes-open resting-state (BF_10_ = 2.759; error % = 0.002). As shown in Table 2 and Figure 4, relative alpha power was reduced in children with ADHD (ADHD-only and ADHD + autism), compared to children without ADHD (ASD-only and typically developing controls), and this difference was larger in the eyes-closed resting-state condition. This result demonstrates how reduced alpha power during eyes-closed resting-state might be an atypicality in brain functioning which is specifically associated with ADHD, and not autism.

### 3.3. Alpha Reactivity

Although the model which best predicted the data on absolute alpha reactivity included the factors “Autism” and “Gender” (BF_model_ = 2.650; error % = 0.662; R^2^ = 0.063), we found anecdotal evidence in support of the absence of a predictive effect of each of these variables on absolute alpha reactivity (Autism: BF_inclusion_ = 0.558; Gender: BF_inclusion_ = 1.012), suggesting that neither Autism nor Gender were strong predictors of absolute alpha reactivity.

Conversely, when investigating relative alpha reactivity, the model including only “full scale IQ” best fit with the data (BF_model_ = 8.784; error % < 0.001; R^2^ = 0.139) and strong evidence was found in support of a predictive effect of IQ on relative alpha reactivity (BF_inclusion_ = 6.370). We followed-up this effect by carrying out a Bayesian correlation analysis. Strong evidence was found in support of the presence of a positive correlation between IQ and relative alpha reactivity (Pearson’s r = 0.464; BF10 = 12.675; 95% Credible Interval = [0.158–0.668]). More specifically, as evidenced in Figure 5, relative alpha reactivity was higher in children with higher IQ.

Due to the presence of differences on IQ between children with ADHD-only and typically developing controls, and between children with ADHD + autism and controls (see Table 1), we investigated if the correlation between IQ and relative alpha reactivity was anyhow confounded by these between-groups differences. We therefore carried out a Bayesian linear regression to investigate if ADHD and autism were predictors of relative alpha reactivity, in addition to IQ. While we found strong evidence in support of IQ as a predictor of relative alpha reactivity (BF = 6.773), we did not find evidence in support of a predictive effect of ADHD or autism (see Table 3).

## 4. Discussion

The present study aimed to investigate electro-physiological profiles of children with or without ADHD and autism, by comparing alpha power and reactivity during eyes-closed and eyes-open resting-state. We found that both absolute and relative alpha power increased over occipital brain areas during eyes-closed compared to eyes-open resting-state. This has been conceptualized and referred to as “alpha reactivity”, and it might be a measure of adaptation of brain activity in different resting-state conditions. Different indices of atypical brain function in relation with ADHD and autism have been found in the present study. More specifically, we found reduced absolute alpha power in children with autism and reduced relative alpha power, particularly during eyes-closed resting-state, in children with ADHD. Children with co-occurring ADHD and autism displayed the same deficits we separately found associated with autism (reduced absolute alpha power) and ADHD (reduced relative alpha, especially at eyes-closed). Our study and findings from Newson and Thiagarajan [25] might therefore indicate that investigating eyes-closed resting-state might be a more reliable method for identifying atypical alpha oscillatory patterns in people with ADHD, who are more likely to display reduced dominant effect of alpha activity in comparison to other frequency bands, during this condition. Conversely, we found reduced absolute alpha power in autism; previous evidence has shown that increased alpha activity at rest is associated with reduced sensitivity to external information, achieved by gating incoming sensory signals [35]. This might indicate that both children with ADHD and autism might be less able to gate incoming information at rest and actively inhibit and reduce sensitivity to sensory information, however children with ADHD might show this difficulty especially during eyes-closed resting-state.

Our findings might have some implications for future research studies investigating arousal in children with ADHD and autism. Reduced relative alpha power in ADHD might be a sign of reduced functional organization of cortical networks during resting-state, especially during eyes-closed, while reduced absolute alpha power in autism might be interpreted as an index of increased sensitivity to sensory information. In a recent study (manuscript in preparation [36]), we have investigated measures of heart rate variability as indices of autonomic arousal, in ADHD and autism. We found that children with ADHD displayed indices of hypo-arousal, especially during less engaging and passive situations, including short periods of resting-state, while children with autism displayed signs of hyper-arousal across different tasks (including resting-state periods). Further research is therefore needed to highlight if different measures, including alpha, electro-dermal activity and heart rate variability, reflect or are differently associated with condition-specific symptoms, such as restlessness during less engaging situations in ADHD, or focused attention on specific aspects of the world in autism.

Unexpectedly, despite the strong evidence found for reduced relative alpha power during eyes-closed resting-state in children with ADHD, we could not replicate previous findings of reduced alpha reactivity in ADHD [17], but we found that children with lower IQ had reduced relative alpha reactivity. Although we might have been underpowered to identify this effect in participants with ADHD, these findings suggest that ADHD might be associated with broad alterations in brain functioning during resting-state, resulting in atypically reduced alpha oscillatory activity in comparison to other frequency bands, during eyes-open and eyes-closed. Conversely, reduced IQ seems to be associated with a specific impairment in the ability to regulate brain states when switching from eyes-open to eyes-closed resting-state. Since resting-state alpha power has been shown to predict performance during cognitive tasks (see [37] for a review), it would be interesting to design future studies and further test the task-to-rest and rest-to-task transitions in these conditions. Similarly, further work is needed to understand the clinical and daily life implications of atypicalities in brain mechanisms supporting arousal regulation, including factors such as sleep, attention and learning, which might be differently associated with ADHD and autism.

There are several limitations in the present study. Data were collected in quite a small sample and during quite short resting-state conditions; we maximized power to investigate effects specific to ADHD and to autism by using Bayesian statistics. Future studies should investigate comorbidity between these conditions and focus on observing dynamic changes in oscillatory patterns over longer periods of resting-state, to determine whether individuals with different conditions differ on these. It is also important to consider that a cognitive task preceded the resting-state session in the present study, and this might have indirectly influenced our resting-state findings.

Despite these limitations, the present study has demonstrated how atypical resting-state oscillatory patterns of alpha activity might be differently associated with ADHD and autism based on the investigated resting-state condition (i.e., eyes-open and eyes-closed). In order to promote collaborations between researchers and research teams, and with the long-term aim of replicating these findings and advancing our knowledge of patterns of electrophysiological oscillatory activity during resting-state in ADHD, we would like to ask individual researchers and research teams to share their EEG data collected during resting-state in individuals with different psychological, psychiatric and neurodevelopmental conditions, including ADHD and autism. In this way, we would be able to create a publicly available repository of resting-state EEG data for further analysis. Full details about this can be found at the following link: https://osf.io/azkhs/, under the section “Files”.

## Figures and Tables

**Figure 1 brainsci-10-00272-f001:**
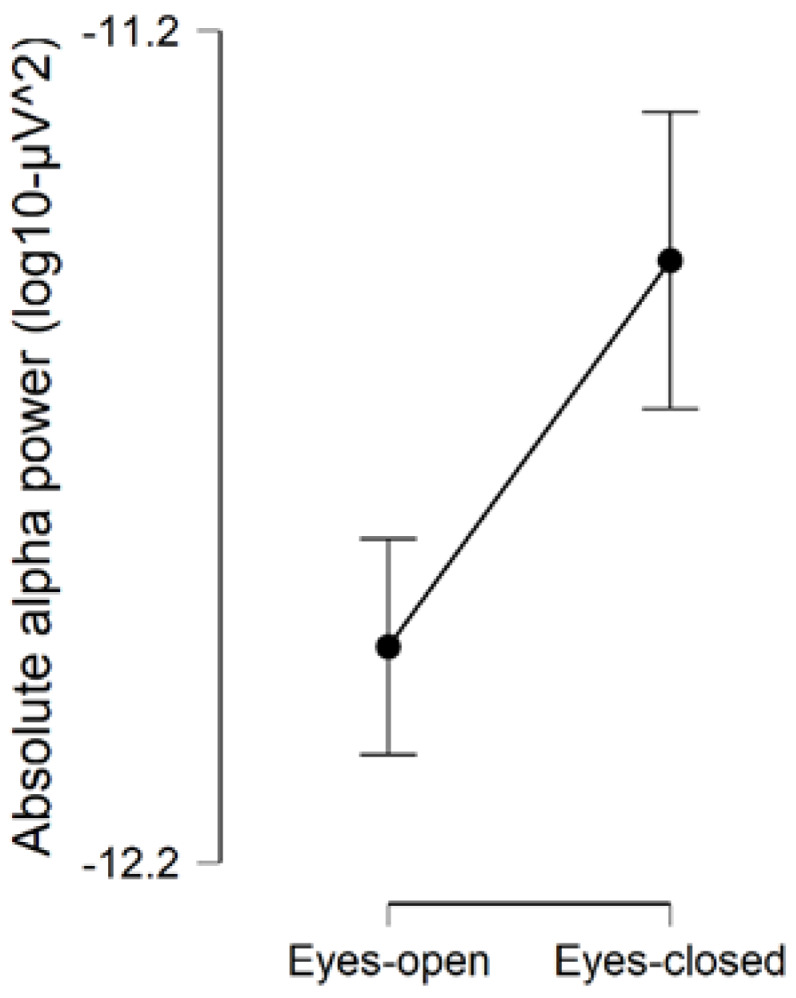
Comparison of absolute alpha power during eyes-open and eyes-closed resting-state conditions. Error bars are 95% credible intervals.

**Figure 2 brainsci-10-00272-f002:**
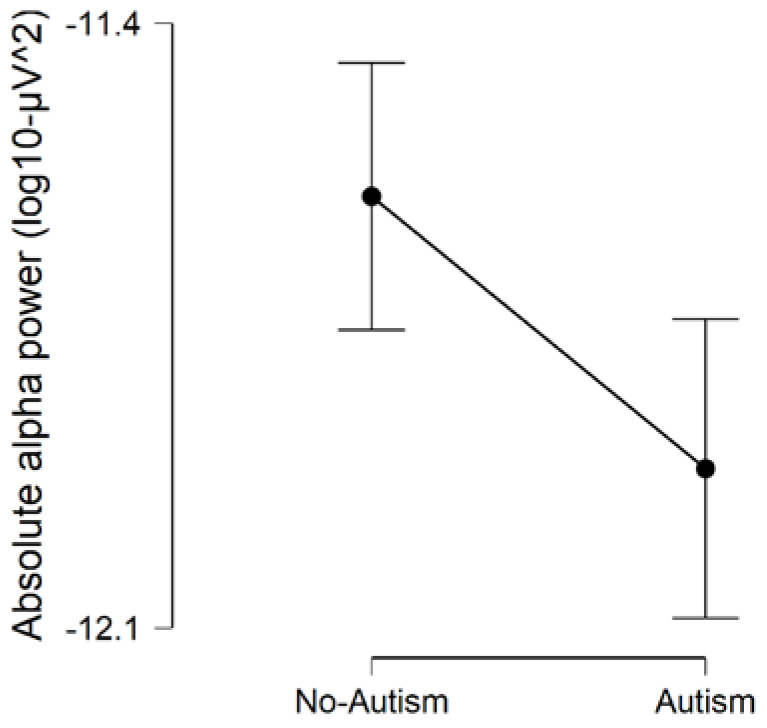
Comparison of absolute alpha power between children with and without autism. Error bars are 95% credible intervals.

**Figure 3 brainsci-10-00272-f003:**
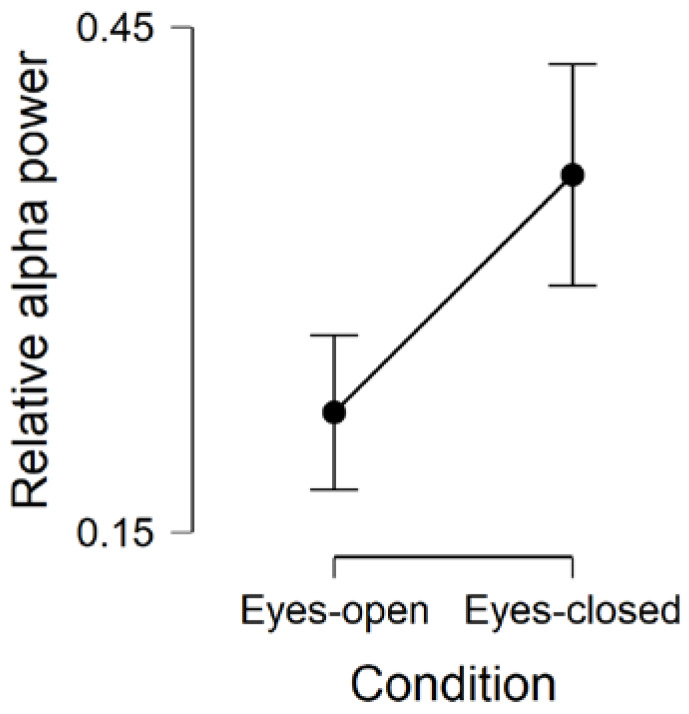
Comparison of relative alpha power during eyes-open and eyes-closed resting-state conditions. Error bars are 95% credible intervals.

**Figure 4 brainsci-10-00272-f004:**
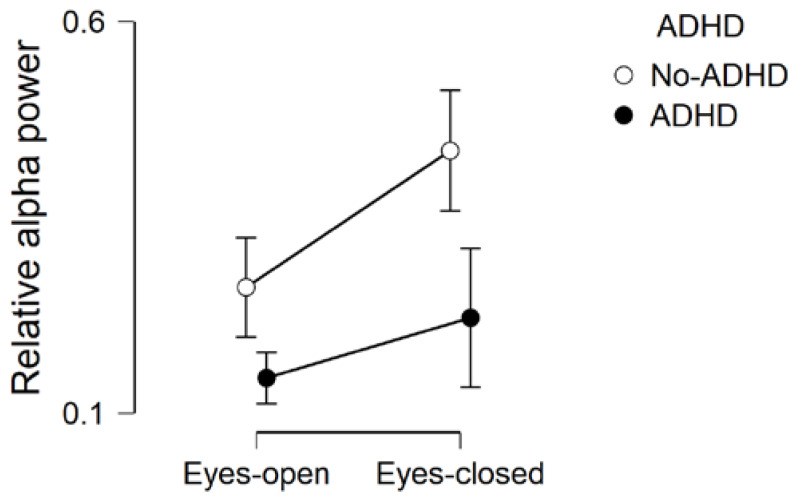
Comparison of relative alpha power during eyes-open and eyes-closed, across children with and without attention deficit hyperactivity disorder (ADHD). Error bars are 95% credible intervals.

**Figure 5 brainsci-10-00272-f005:**
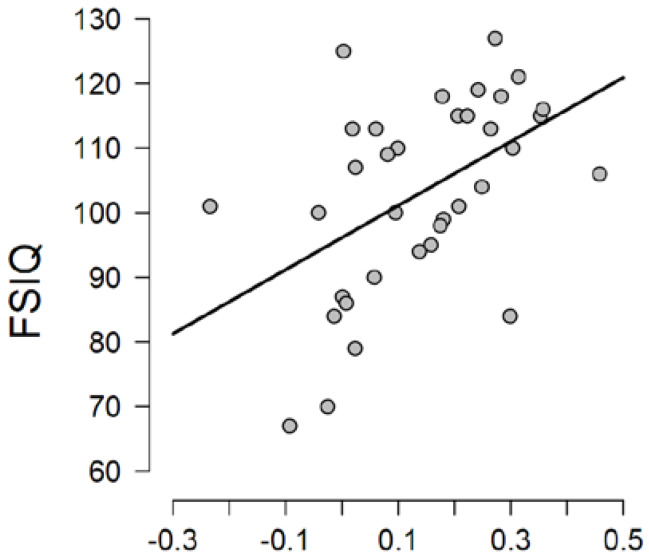
Correlation plot between full scale IQ (FSIQ) and relative alpha reactivity.

**Table 1 brainsci-10-00272-t001:** Socio-demographic and clinical characteristics of the sample included in the present study.

Demographic and Clinical Variables Mean [SD]	Typically Developing Controls	ADHD-Only	Autism-Only	ADHD + Autism	Group Comparisons
N (females)	20 (3)	9 (5)	6 (0)	8 (0)	-
Mean age (years)	11.93 [1.78]	12.09 [2.25]	12.56 [2.24]	13.12 [1.78]	None
Social Communication Questionnaire (total score)	3.35 [3.74]	13.75 [6.18]	20.50 [9.09]	26.25 [5.68]	ADHD + autism > ADHD-only; ADHD-only, ADHD + autism and autism-only > TD
Conners’ Rating Scales (total score)	41.12 [16.35]	80.78 [13.08]	64.17 [12.51]	78.75 [3.28]	ADHD-only, ADHD + autism and autism-only > TD
Full Scale IQ	111.19 [9.62]	93.00 [17.26]	107.17 [8.89]	89.86 [15.82]	ADHD-only < TD; ADHD + autism < TD

Mean values are reported for each group, with standard deviations in parentheses.

**Table 2 brainsci-10-00272-t002:** Comparison of relative alpha power during eyes-open and eyes-closed resting-state conditions, between children with and without ADHD.

Condition	Group	Mean Relative Alpha Power	SD	95 % Credible Interval	
				Lower	Upper
Eyes-open	No-ADHD	0.258	0.152	0.197	0.319
	ADHD	0.162	0.080	0.121	0.203
Eyes-closed	No-ADHD	0.442	0.187	0.367	0.518
	ADHD	0.261	0.154	0.181	0.340

**Table 3 brainsci-10-00272-t003:** Results of the Bayesian linear regression on relative alpha reactivity, investigating the predictive effect of IQ, ADHD-, and autism-diagnosis.

Models	Bayes Factor	R²
a. IQ	6.773	0.215
b. ADHD	0.601	0.105
c. Autism	0.125	0.006
d. ADHD + autism	0.226	0.105
e. IQ + ADHD + autism	0.626	0.222
f. IQ + ADHD	1.684	0.218
g. IQ + autism	1.676	0.218

Bayes Factor and R^2^ are reported for each of the regression models investigated, which included one predictive factor (models **a**.–**c**.) or a combination of several predictive factors (models **d**.–**g**.).

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
