# Peer review of "Atypical Electrophysiological Indices of Eyes-Open and Eyes-Closed Resting-State in Children and Adolescents with ADHD and Autism"

_brainsci, 2020, doi:10.3390/brainsci10050272_

Round 1

Reviewer 1 Report

The authors have investigated the "electrophysiological indices of resting-state EEG in children and adolescents with ADHD and co-morbid autism".

The paper is well-written but I am afraid that the topic lacks originality (see the works from Koehler and colleagues 2008 (for ADHD), Dawson et al 1995 and Coben et al 2008 (for ASD).

Moreover, I have the following concerns:

1)  The introduction is very long and does not go to the point. Some pieces of the literature lack, especially with respect to the studies already performed on children with ASD and ADHD using resting-state EEG (see the papers above). Also, there are some sentences that should be moved to the discussion or results.

2)The study design does not address the main hypothesis stated in the title since the group with comorbid ADHD-ASD is very small (8 subjects) and the authors could not investigate specific EEG parameter for this population (Pag. 4 lines 151-152)

3) The methodology of FFT is unclear to me.  FFT was performed on the average of all the 2-seconds long consecutive epochs? If that is the case, the authors should be very cautious in interpreting their results based on the analysis of such a short period of time. Also, the authors should mention how many epochs were selected on average for each group.

4) No correction for multiple comparisons is reported.

5) I would encourage to perform the same analysis on other frequency bands (i.e. delta to beta), in order to have a better understanding of the neurophysiological resting-state status of these patients.

6) The authors should report if they found significant differences in the clinical characteristics of ADHD + Autism, ADHD-only, and Autism-only group. Also, it is unclear if they have considered the ADHD+Autism subjects belonging to both ADHD-only and Autism-only groups in the final analysis. This would be debatable since they represent a different kind of patients population.

7) Regression model: It is unclear if other predictors were evaluated other than “IQ”, ADHD and Autism (i.e. Communication Questionnaire, Conner’s rating scale).

8) Discussion: “This is in line with previous studies (see [26] for a review) and demonstrates how during eyes-closed resting-state the expected dominant effect of alpha activity, in comparison to other frequency bands, is likely to be reduced in the presence of ADHD symptoms.” This is not true since that specified review reports many inconsistencies in the studies which were evaluated, and the most common result for alpha activity in ADHD was “no difference” between groups.

9)  Discussion: Pages 10 lines 341-353. And Page 11 lines 366-370. Speculations unrelated to results.

Author Response

Dear editors, dear reviewer,

We would like to thank you for the time you spent reading our manuscript and providing feedback on it. We have now revised the manuscript according to the reviewer’s comments; and we are now presenting our comments in response to the reviewer for each point.

Reviewer 1

The authors have investigated the "electrophysiological indices of resting-state EEG in children and adolescents with ADHD and co-morbid autism".

The paper is well-written but I am afraid that the topic lacks originality (see the works from Koehler and colleagues 2008 (for ADHD), Dawson et al 1995 and Coben et al 2008 (for ASD).

  • We would like to thank Reviewer 1 for reading our manuscript and providing their positive feedback about how the study is presented. We agree with them that investigating resting-state EEG profiles is not a completely novel and original approach, so that extensive reviews on this topic have been already published (see, for example, Newson & Thiagarajan, 2018; cited in our manuscript). Our study originated from the fact that only few previous studies have focused on mixed samples of children with ADHD and Autism (we have cited those who did in the manuscript, e.g., Shephard et al., 2018) and at the same time analysed EEG oscillatory patterns in different resting-state conditions (i.e., eyes-closed and eyes-open). For example, Shephard et al. (2018) only had an eyes-open resting-state condition, while Fonseca et al. (2013) analysed the difference in alpha power between eyes-open and eyes-closed resting-state, but only in children with ADHD.

Moreover, I have the following concerns:

1) The introduction is very long and does not go to the point. Some pieces of the literature lack, especially with respect to the studies already performed on children with ASD and ADHD using resting-state EEG (see the papers above). Also, there are some sentences that should be moved to the discussion or results. 

  • We thank the reviewer for pointing out that the introduction did not go straight to the point, we have reduced it now and made it more focused on the research questions of the study (page 2, lines 35-60). 
  • We have not focussed on specific papers or studies which investigated resting-state EEG in ADHD or Autism, since most of the literature on this topic has been summarised in the review by Newson & Thiagarajan (2018). The papers mentioned by reviewer 1 (Coben et al., 2008; Dawson et al., 1995; and Koehler et al., 2008) are in fact included in the main summaries for ADHD and Autism (Newson & Thiagarajan, 2008; see Figure 7, page 10).
  • The comment 'there are some sentences that should be moved to the discussion or results' might be related to the last section of the Introduction, which briefly summarises the results and conclusions of the study: we have added this section as indicated in the Instruction for Authors (https://www.mdpi.com/journal/brainsci/instructions): ' Introduction: [..] briefly mention the main aim of the work and highlight the main conclusions'.

2) The study design does not address the main hypothesis stated in the title since the group with comorbid ADHD-ASD is very small (8 subjects) and the authors could not investigate specific EEG parameter for this population (Pag. 4 lines 151-152). 

  • We are aware that sample size of the present study was not very large and only few children were part of the comorbid group.
  • We have realised that the sentence originally at page 4 (lines 151-152) might be misleading, therefore we have now changed it (see page 4, lines 137-139).

3) The methodology of FFT is unclear to me. FFT was performed on the average of all the 2-seconds long consecutive epochs? If that is the case, the authors should be very cautious in interpreting their results based on the analysis of such a short period of time. Also, the authors should mention how many epochs were selected on average for each group.

  • Fast Fourier Transformation (FFT) was calculated for each epoch, and then averaged. This is what has been commonly done in previous studies (e.g., Koehler et al., 2008). The length of the epochs (i.e., 2 seconds) is in line with previously published papers and guidelines (see, for example, Babiloni et al., 2020; https://doi.org/10.1016/j.clinph.2019.06.234). Moreover, since we have decided to specifically focus on the alpha frequency band, a 2-seconds-long temporal window is considered appropriate to accurately estimate oscillatory power in this frequency band.

4) No correction for multiple comparisons is reported. 

  • We thank Reviewer 1 for this comment; we agree that if we had taken a traditional frequentist approach, we would have to correct for multiple comparisons. This is partly why we had decided to use Bayesian statistics, i.e., investigating the probability of the presence of main effects of ADHD or Autism (and, therefore, the probability of the presence of differences between those with and without a specific condition). Using Bayesian statistics, instead of a traditional frequentist approach, has allowed us to compensate for the reduced sample size and power of the study.

5) I would encourage to perform the same analysis on other frequency bands (i.e. delta to beta), in order to have a better understanding of the neurophysiological resting-state status of these patients. 

  • Although we appreciate the advice given by Reviewer 1, we would like to point out how we have specified the reasons why we have decided to focus on alpha, in the Introduction (see page 2, lines 48-60). More specifically, since none of the original hypotheses of the study were associated with other frequency bands than alpha, we have decided to only circumscribe the analysis to this frequency band.

6) The authors should report if they found significant differences in the clinical characteristics of ADHD + Autism, ADHD-only, and Autism-only group. Also, it is unclear if they have considered the ADHD+Autism subjects belonging to both ADHD-only and Autism-only groups in the final analysis. This would be debatable since they represent a different kind of patients population.

  • We have reported any between-groups differences on clinical measures (including SCQ scores, Conners’ scores and IQ) in Table 1, page 4, line 141. I think Reviewer 1 refers to the Bayesian linear regression when they mentioned the ‘final analysis’. In this case (see Table 3, page 8, line 284), the label ‘ADHD+Autism’ does not refer to the group of children with comorbid ADHD+Autism, but to the regression model which included both ADHD and Autism as predictive factors. We have added a table caption to explain these factors in more details (page 9, lines 285-286).

7) Regression model: It is unclear if other predictors were evaluated other than “IQ”, ADHD and Autism (i.e. Communication Questionnaire, Conner’s rating scale).

  • Other predictors have not been investigated in the regression analysis, besides IQ, ADHD and Autism. This analysis was carried out to follow-up of the results presented at page 8, lines 267-269, and, more specifically, to investigate if there was a confounding effect of ADHD or Autism on the relationship between IQ and relative alpha reactivity.

8) Discussion: “This is in line with previous studies (see [26] for a review) and demonstrates how during eyes-closed resting-state the expected dominant effect of alpha activity, in comparison to other frequency bands, is likely to be reduced in the presence of ADHD symptoms.” This is not true since that specified review reports many inconsistencies in the studies which were evaluated, and the most common result for alpha activity in ADHD was “no difference” between groups.

  • We agree with Reviewer 1 in saying that many inconsistencies were reported by different studies; however, as reported in Table SM4 of the original paper (https://www.frontiersin.org/articles/10.3389/fnhum.2018.00521/full#supplementary-material), in most of the studies children with ADHD have been found to show reduced relative alpha during EC (and this is what we have found in our study). Our study and the review by Newson & Thiagarajan (2008) might then indicate that investigating eyes-closed resting-state might be a more reliable method for identifying atypical alpha oscillatory patterns in ADHD (we have added this to the discussion, page 9, lines 296-300).

9)  Discussion: Pages 10 lines 341-353. And Page 11 lines 366-370. Speculations unrelated to results.

  • Although we are aware that it would be interesting to consider how our results might be associated with sleep problems in ADHD and Autism, and how they might relate with general attentional differences, learning, social and cognitive development we agree with Reviewer 1; since we didn’t measure sleep difficulties in the sample, these sentences might be interpreted as speculations, therefore we have now deleted them and substituted with the sentence at page 10, lines 327-331).

Reviewer 2 Report

This paper presents a research on brain functioning and responsitibility for ADHD and Autism. They compared the alpha power for the cases of eye-open and eye-closed cases. The proved their argument through various comparisons of alpha powers in eye-open and eye-closed cases. P9.L292 counfounded --> confounded

Author Response

Dear editors, dear reviewer,

We would like to thank you for the time you spent reading our manuscript and providing feedback on it. We have now revised the manuscript according to the reviewer’s comments; and we are now presenting our comments in response to the reviewer for each point.

Reviewer 2

This paper presents a research on brain functioning and responsitibility for ADHD and Autism. They compared the alpha power for the cases of eye-open and eye-closed cases. The proved their argument through various comparisons of alpha powers in eye-open and eye-closed cases. P9.L292 counfounded --> confounded

  • We would like to thank Reviewer 2 for reading our manuscript and providing their positive feedback. We are glad that they appreciated how previous literature is presented in the introduction, and how our study fits with it. We are also happy they appreciated the use of Bayesian statistics as the primary methods of analysis and how we presented our results and conclusions. 
  • Thanks for pointing out the misspelledword 'counfounded', we have now corrected it (page 8, line 279).

Round 2

Reviewer 1 Report

The authors have answered most of my concerns thoroughly and the manuscript is now significantly improved.

My last minor concern is that I am not fully convinced of the originality of their work, considering that they were not able to study the population of ADHD and co-morbid autism. A further improvement in the introduction (which is still very long) with a clear statement about the main scientific hypothesis, the motivations for the need of another study on this subject, considering the existing literature, and the reasons explaining their final methodology, may eventually resolve this problem.

Also, despite their claims,  the authors on page 2 lines 48-60 do not explain why they focused on alpha activity only (no mention of "alpha" at all). Thus I still do not understand why they have not studied other frequency bands, to provide additional insights about this topic.

Author Response

Dear reviewer and editors, we would like to thank you again for having provided feedback on the revised draft of our manuscript. Here you can find our comments, point by point, to the Round 2 of reviewers’ comments.

The authors have answered most of my concerns thoroughly and the manuscript is now significantly improved.

  • We are happy to hear that Reviewer 1 think the manuscript has improved from the original submission.

My last minor concern is that I am not fully convinced of the originality of their work, considering that they were not able to study the population of ADHD and co-morbid autism. A further improvement in the introduction (which is still very long) with a clear statement about the main scientific hypothesis, the motivations for the need of another study on this subject, considering the existing literature, and the reasons explaining their final methodology, may eventually resolve this problem.

  • As highlighted in the first round of comments, we agree with Reviewer 1 in saying that investigating resting-state EEG profiles is not a fully original approach. We have slightly reduced the introduction, to specifically highlight the reasons behind carrying out the present study in addition to the wide existing literature, as suggested by Reviewer 1.
  • Moreover, since we think our work adds some original content to the literature review published by Newson & Thiagarajan (2018), we have specified how our study investigated condition-specific effects of ADHD and Autism in a mixed sample of children with ADHD and/or Autism, and analysed EEG alpha oscillatory patterns in different resting-state conditions (i.e., eyes-closed and eyes-open), which no previous study has done before (see page 3, lines 93-96).
  • Although we do not completely agree with the statement 'they were not able to study the population of ADHD and co-morbid autism', we understand the point. In fact, although we did not find any specific effects associated with the co-morbid group, we found that this group displayed the same deficits separately found associated with Autism (reduced absolute alpha) and ADHD (reduced relative alpha, especially at eyes-closed) (see page 9, lines 293-295).

Also, despite their claims, the authors on page 2 lines 48-60 do not explain why they focused on alpha activity only (no mention of "alpha" at all). Thus I still do not understand why they have not studied other frequency bands, to provide additional insights about this topic.

  • We thank Reviewer 1 for the comment. We have therefore decided to follow their advice and specify the reasons why we have only focused on alpha and not on other frequency bands (see page 2, lines 46-71).